**Data Availability Statement:** All relevant data are within the paper and Supporting Information files.

**Funding:** This study was supported by National Ministry of Science & Technology (2019YFC1710202), the Shandong Science &

# Global risk factor analysis of myopia onset in children: A systematic review and meta-analysis

Mingkun Yu[1]☯, Yuanyuan Hu[2,3,4]☯, Mei Han[5], Jiawei Song[6], Ziyun Wu[6], Zihang Xu[6], Yi Liu[7], Zhen Shao[7], Guoyong Liu[6], Zhipeng Yang[6], Hongsheng Bi [2,3,4]*

1 The First Clinical Medical College, Shandong University of Traditional Chinese Medicine, Jinan, China, 2 Shandong Provincial Key Laboratory of Integrated Traditional Chinese and Western Medicine for Prevention and Therapy of Ocular Diseases, Jinan, China, 3 Key Laboratory of Integrated Traditional Chinese and Western Medicine for Prevention and Therapy of Ocular Diseases in Universities of Shandong, Jinan, China, 4 Eye Institute of Shandong University of Traditional Chinese Medicine, Jinan, China, 5 Beijing University of Chinese Medicine, Beijing, China, 6 Ophthalmology & Optometry Medical School, Shandong University of Traditional Chinese Medicine, Jinan, China, 7 Affiliated Eye Hospital of Shandong University of Traditional Chinese Medicine, Jinan, China

☯ These authors contributed equally to this work.
* hongshengbi@126.com

## Abstract

### Introduction

This work aimed to comprehensively assess the risk factors affecting myopia in children to develop more effective prevention and treatment strategies. To this end, data from database were employed to assess the relationship between the incidence of myopia and its risk factors.

### Methods

We searched eight databases online in June 2022. Cohort studies were included that measured the connection between risk factors and myopia. Eligibility was not restricted by language. The Newcastle–Ottawa Scale (NOS) was used to measure the risk of bias and conducted GRADE evaluation to determine the certainty of evidence. Potential risk factors with positive or negative results were seen. Inplasy Registration: https://inplasy.com/inplasy-2022-4-0109/.

### Results

Evidence that risk factors for myopia are mixed, comprising both positive (20) and null (17) findings. In 19 cohort studies on 3578 children, girls were more likely to develop myopia (RR: 1.28 [1.22–1.35]). Myopia can occur at any age, from early childhood to late adulthood. Children whose parents had myopia were more likely to develop myopia. Longer outdoor activities time (RR: 0.97 [0.95–0.98]) and less near-work time (RR: 1.05 [1.02–1.07]) appeared to be significantly decrease the incidence of myopia. Children with lower SE, longer AL, a lower magnitude of positive relative accommodation, worse presenting visual acuity, deeper anterior chamber, and thinner crystalline lens may be related to myopia onset.

Technology Department (2019GSF108252), Shandong Department of Finance (YXH2019ZXY001) and Shandong Health Commission (202107020948). The funders had no role in study design, data collection and analysis, decision to publish, or preparation of the manuscript.

**Competing interests:** The authors have declared that no competing interests exist.

**Abbreviations:** AL, Axial length; CNKI, Chinese National Knowledge Infrastructure Database; $I^2$, Heterogeneity; IQ, Intelligence quotient; NOS, The Newcastle-Ottawa Scale; OR, Odds ratio; RR, Risk ratio; SE, Spherical equivalent; AL/CR ratio, Axial length-corneal radius; AC, anterior chamber; AC/A ratio, accommodative convergence to accommodation; ACD, Anterior chamber depth.

The burden of myopia in underprivileged countries is higher than in developed countries (RR: 5.28 [2.06–13.48]). The quality of evidence for the evaluated factors was moderate to low or very low.

## Conclusions

Genetic factors, environmental factors (such as excessive use of electronic products, and poor study habits) and lifestyle factors (such as lack of outdoor activities, poor nutrition, etc.) are the main risk factors for myopia in children. Myopia prevention strategies should be designed based on environmental factors, gender, parental myopia and eye indicators in order to explore a lifestyle that is more conducive to the eye health of children.

## Introduction

Myopia is one of the most common visual problems in the world [1], has become one of the most common eye diseases in the world, is also one of the main causes of blindness and visual impairment [2,3]. The incidence of myopia in children is increasing rapidly worldwide, becoming a serious public health problem [4,5]. With a global population of 6.4 billion, about 31.3 percent are diagnosed with myopia [6,7]. It is estimated that cases of myopia will increase to 4.76 billion people (49.8% of the global population) by 2050. Myopia is a significant risk factor for other vision-threatening conditions such as glaucoma, myopic macular degeneration, and retinal detachment. As the incidence of myopia increases, the risk ratio (RR) for these diseases increases dramatically [8,9]. It has been reported that both environmental and genetic factors contribute to the cause of myopia. Studies have shown that the incidence of myopia is genetic, but also has a strong family aggregation, the incidence between brothers and sisters can reach 2.09 ~ 3.86 [10,11]. Some people also say that genetic factors are not the main factors in the process of myopia, but the influence of environmental factors can completely eliminate the influence of genetic factors. Changes in environmental factors play a decisive role in the worldwide prevalence of myopia. Although myopia has been widely studied, genetic and environmental theories remain controversial. It was not until the 1970s that research on the basis of vision found that if an animal's retina could not obtain a clear image, the axis of the eye would lengthen, leading to myopia. A new theory, form deprivation, suggests that refractive errors are more closely related to the environment than to genetics. The specific effects of other genetic risk factors remain controversial. The pathogenesis of myopia, mechanical axis growth theory, environmental theory, genetic theory, and form deprivation theory have provided some new ways to discover the factors of myopia. In addition, the differences in myopia rates and influencing factors among children around the world also need to be further discussed [12,13]. Therefore, understanding the global risk factors for childhood myopia and their impact is of great significance for developing effective prevention and intervention measures.

The number of cohort studies exploring risk factors for myopia has increased markedly. The aim of meta-analysis was to comprehensively assess risk factors of myopia onset. There are some differences in the risk factors of myopia in children in different regions, so intervention and management should be carried out according to local conditions. This study provides valuable information for in-depth understanding of the pathogenesis of myopia in children, which can help to develop more accurate prevention and treatment measures to promote the eye health and all-round development of children.

## Materials and methods

### Design

The registered protocol is available at https://inplasy.com/inplasy-2022-4-0109/.

### Research questions

Several research questions were addressed by this scoping review:

1. Which risk factors can predict the onset of myopia?

2. Which strategies can be used to prevent myopia onset?

### Eligibility criteria and outcome measures

An online search of six databases (CNKI, VIP, WanFang, Pubmed, Embase and Cochrane) and two two clinical trials registries was performed in June 2022. A reference search was also conducted for relevant studies. We included cohort studies that evaluated the association between potential factors and myopia onset. All studies on children aged 6–18 years old were included. Thus, studies that reported any potential factors with positive or negative results included: parents factors (refractive status of parents, education of parents, etc.), personal factors (age, sex, ethnicity, intelligence quotient [IQ], school grade, BMI, household income, etc.), indicators of the eyes (axial length [AL], visual acuity, accommodation, spherical equivalent [SE] degree, etc.), environmental factors (socioeconomic status, second-hand smoking environments, etc.), and behavioral factors (near-work time, duration of sleep, outdoor activity time, night-light use, distance viewed, time spent reading, etc.). All possible potential factors related to the incidence of myopia were included in our study. The included studies proposed measures of correlation, which could be 95% confidence intervals or standard deviations, or calculations based on the primary data provided in this article. The Studies regarding the linkage between genes and myopia were not included. In the current study, the definitions of myopia, myopia lifestyles were based on those included studies.

### Data sources and literature searches

Two authors (YY Hu and MK Yu) comprehensively searched published studies from their inception to April 2022: PubMed, the Cochrane Library, EMBASE, Chinese National Knowledge Infrastructure Database (CNKI), VIP Chinese Science and Technique Journals Database, and Wanfang Database. We also searched for two clinical study registration networks (clinical trials.gov and Chinese clinical trial registry). The references of retrieved articles were also screened to identify other potentially relevant articles. The search strategy is displayed in Appendix.

### Data extraction

**Study selection.** Import the retrieved files into NoteExpress and delete duplicate data. Two researchers (YY Hu and MK Yu) conducted the preliminary screening of the literature independently by reading the titles and abstracts. Afterwards, a full search was performed on all potentially satisfying papers, and two researchers were assessed for eligibility. Any discrepancies were discussed with a third researcher. Then, the full texts of potentially eligible studies were retrieved and assessed for eligibility by the two authors. Any disagreement between them on the eligibility of particular research was resolved through discussions with a third reviewer (Hongsheng Bi).

**Data extraction and management.** The full texts of all relevant studies included in this process were retrieved and screened independently by the same reviewers. Data extraction, including authors' information (such as name and number), characteristics of the studies (such as publication date, location, region of research, the latitude of the study location), information related to the study design, definitions (such as myopia and outdoor activity), adjustment for confounding factors, and outcomes (ORs, HRs, RRs, 95% CI, and standard error), was conducted by two groups of reviewers (group 1: ZH Xu and ZY Wu; group 2: Z Shao and Y Liu) using an established data extraction form. Any discrepancies were discussed with a third researcher. The influencing factors of myopia onset were extracted with an open structure. This checklist was continuously updated as the group discussed if new factors were discovered, and finally this checklist was used as a template for documenting the various elements to be included in this project.

## Quality assessment (risk of bias)

The quality of the included cohort studies was assessed using the Newcastle–Ottawa Scale (NOS) checklist [14], which consists of eight sections and divides the studies on a scale of zero to nigh, indicating poor to high quality, respectively.

## Data analysis

Stata (Version 12.0, Stata Corporation, College Station, TX, USA) was used to analyze the data. We conducted meta-analyses of pooled study outcomes only if the studies provided adequate data and did not have clinical heterogeneity. Otherwise, the results were presented as a narrative summary. The summary measure used the RR with a 95% CI to determine the association between potential factors and myopia. If important information is not provided, the corresponding author will be contacted. Over ten studies were evaluated by funnel plots to assess publication bias.

## Subgroup analysis and sensitivity analysis

A heterogeneity ($I^2$) assessment was performed using Stata. When heterogeneity was not significant ($I^2 < 50\%$), the fixed-effect model was used to process the data. When heterogeneity was significant ($I^2 \geq 50\%$), we used the random-effects model [15]. The potential sources of heterogeneity, namely, cycloplegia or the definition of myopia (S1 Table), could be considered by subgroup analyses. We aimed to analyze the potential sources of heterogeneity by performing prespecified subgroup analyses. We made a sensitivity analysis to assess the robustness of the meta-analysis by excluding more than one risk of high bias or two or more risks of unknown bias. GRADE was referenced to evaluate the quality of evidence.

# Results and discussion

A total of 3,578 studies were screened, and 19 cohort studies [16–34] were included (Fig 1).

## Study characteristics

Total 19 cohort studies [16–34] included 3578 children aged 7–18 years old. Fourteen were cohort studies [16–26,28,29,34] conducted in East Asia, and of which 14 cohorts. [16–26,28,30,31] were followed up in the schools. Among the studies, [16–18,20–34] 18 studies were funded by the public, whereas there was no funding reported by Wang [19]. Sex was not reported in seven studies [20,25–27,30–32]. In the remaining 12 studies [16–19,22–24,28,32–

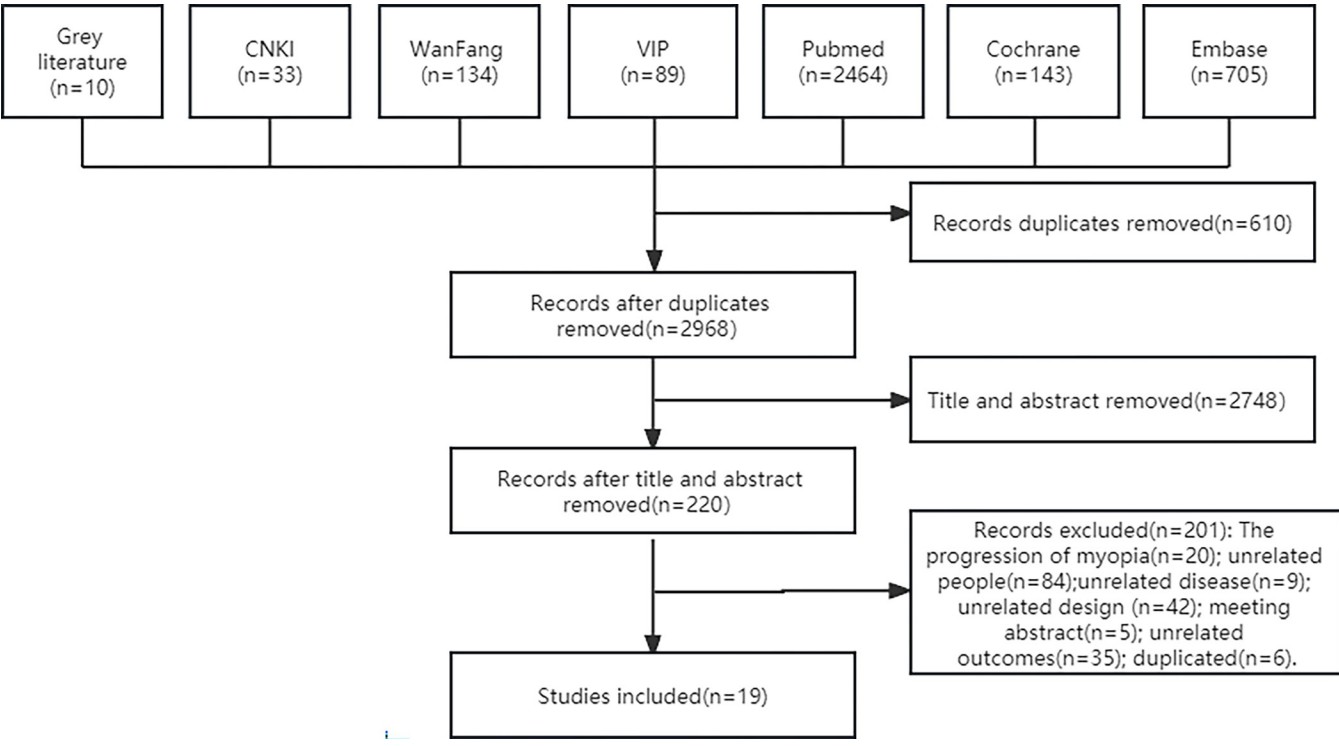

**Fig 1. Study flow diagram.**

34], the proportion of boys to girls was approximately 1:1.3. Related studies were published between 2001 and 2022 (Table 1).

## Risk of bias

Based on NOS checklist (S2 Table), 17 cohort studies [16–28,30,31,33,34] were evaluated as high quality, with 10 studies having a score of 9; [16,18–23,26,27,34] four studies; had a score of 8; [17,25,30,31] and three studies, a score of 7 [24,28,33]. The reviewers considered that two studies (a score of 6) did not adequately follow the cohort (Fig 2).

## The results of included outcomes

Data analysis was performed according to our protocol. Negative results were provided in Table 2.

   **Genetic factors.** Parental dioptric status was reported in these cohort studies [16,20,22,25,27,30,32–34] (myopic versus non-myopic parents). Twelve cohorts [16,20,22,25,27–30,32–34] demonstrated that children who have one parent with myopia may have a higher risk of myopia onset (RR = 1.16; 95% CI: 1.05–1.27; middle certainty of evidence; $I^2$: 63.3%; n = 24,460) than children who have parents without myopia. The study was evaluated as providing a medium range of certainty of evidence (Fig 3A). The funnel plot appeared symmetrical, thus indicating that no serious publication bias existed (P = 0.49; Fig 3). Substantial active connection between children who have two parents with myopia and the possibility of developing myopia (RR: 1.42; 95% CI: 1.26–1.57; $I^2$: 63.6%, n = 24,460; Fig 3). The subgroup analysis suggested children who have one parent with myopia (SE $\leq$ -0.75D with cycloplegia: RR: 1.62; 95% CI: 1.33–1.91, $I^2$ = 0%, 4 cohorts; Fig 3) or two parents with myopia (SE $\leq$ -0.5D

**Table 1. Characteristics of the included 19 cohort studies.**

| Study ID | Funding | Study location | Follow-up setting | Follow-up time | Study Population(Age (Mean/Range)) | Sample | Whether to Cycloplegic | Statistical analysis | Definition of Myopia (D) | Factors collected |
|---|---|---|---|---|---|---|---|---|---|---|
| Li SM2022 [16] | Public | East Asia | School | 5 years | Chinese school non-myopia children from grade 1 to grade 6(7.2 ±0.3) | 2835 | Yes | Multivariable logistic regression | SE≤−0.5 D | 1*,2,5,9,10,16,23–25 |
| Huang L2021 [17] | Public | East Asia | School | 3 years | Non-myopia children | 26611 | Yes | Multivariable logistic regression | SE≤−0.5 D | 1–5*,6,7,26* |
| Jiang D2021 [18] | Public | East Asia | School | 2.5 years | Grade 1–3 non-myopia children(7.29±0.91) | 1388 | No | Multivariable COX Regression | SE ≤ − 1.0 | 1,2,6,7,9,10,21 |
| Wang BN2021 [19] | NR | East Asia | School | 1 years | Non-myopia children (11.6±1.8) | 2015 | No | Multivariable logistic regression | NR | 2,6,7,27 |
| Wong YL2021 [20] | Public | East Asia | School | 2 years | Non-myopia children (7.8±0.7) | 1066 | Yes | Multivariable logistic regression | SE≤−0.5 D | 1,2,6,7,18,27–30 |
| Qi LS2019 [21] | Public | East Asia | School | 3 years | Non-myopia children (15.5 ± 0.6) | 522 | Yes | Multivariable logistic regression | SE≤−0.5 D | 5–7,10,12–14,31 |
| Ma Y2018 [22] | Public | East Asia | School | 4 years | Non-myopia children (8.1±1.1) | 1856 | Yes | Multivariable logistic regression | SE≤−0.5 D | 1*,2*,5–7,10* |
| Ma Y2018 [23] | Public | East Asia | School | 2 years | Non-myopia children | 770 | Yes | Multivariable logistic regression | SE≤−0.5 D | 2,5,9,10,28,33,36 |
| Wang SK2018 [24] | Public | East Asia | School | 6 years | Non-myopia children | 2599 | No | Multivariable logistic regression | SE≤−0.5 D | 2,9,10,15,23,25 |
| Tsai DC2016 [25] | Public | East Asia | School | 1 years | Non-myopia children (7–8) | 11590 | Yes | Multivariable COX Regression | SE≤−0.5 D | 5,6,10,12,22,32 |
| Chua SY2016 [26] | Public | East Asia | School | 3 years | Pregnant women and their children (birth cohort) | 1236 | Yes | Multivariable logistic regression | SE≤−0.5 D | 17 |
| Zadnik K2016 [27] | Public | Non-East Asia | Other | 31 years | Non-myopia children (6–11) | 4512 | Yes | Multivariable logistic regression | SE ≤ -0.75D | 1,2*,5,8*,16,20,33–35 |
| Ma YY2016 [28] | Public | East Asia | School | 4 years | Non-myopia children (8.05(8.00 to 8.11)) | 1639 | Yes | Multivariable logistic regression | SE≤−0.5 D | 1,2,5–7,10,13,18 |
| Chua SY2015 [29] | Public | East Asia | Other | 3 years | Non-myopia children (birth cohort) | 1086 | Yes | Multivariable logistic regression | SE≤−0.5 D | 1*,2*,3*,5–7,8*,11* |
| French AN2014 [30] | Public | East Asia | School | 2 years | Non-myopia children (younger cohort and older cohort) | 2103 | Yes | Multivariable logistic regression | SE≤−0.5 D | 1*,2*,5,6,7 |
| French AN2013 [31] | Public | Non-East Asia | School | 1 years | Non-myopia children (younger cohort and older cohort) | 4118 | Yes | Multivariable logistic regression | SE≤−0.5 D | 1*,2, 8 |
| Jones-Jordan LA2010 [32] | Public | Non-East Asia | Other | 6 years | c(6–14) | 2158 | Yes | Multivariable COX Regression | SE ≤ -0.75D | 5, 7, 9 |
| Jones LA2007 [33] | Public | Non-East Asia | School | 12 years | Non-myopia children | 514 | Yes | Multivariable logistic regression | SE ≤ -0.75D | 5–7, 9, 10, 13 |

(*Continued*)

**Table 1.** (Continued)

| Study ID | Funding | Study location | Follow-up setting | Follow-up time | Study Population(Age (Mean/Range)) | Sample | Whether to Cycloplegic | Statistical analysis | Definition of Myopia (D) | Factors collected |
|---|---|---|---|---|---|---|---|---|---|---|
| Saw SM2006 [34] | Public | East Asia | School | 3 years | Non-myopia children (7–9) | 994 | Yes | Multivariable logistic regression | SE ≤ -0.75D | 1,2,4,5,18*,19 |

1.Age; 2.Gender; 3.Level of parental education; 4.Household income; 5.Parental myopia; 6.Near work time; 7.Outdoor activity time; 8.Ethnicity; 9.Axial length; 10. Spherical equivalent; 11.Height; 12.Distance of reading; 13.Time of reading; 14.Duration of sleeping; 15.Weight; 16.Crystalline lens power; 17.Second-hand smoking; 18. School; 19.Intelligence quotient; 20.Astigmatism magnitude; 21. Body Mass Index;22.Region of residence;23.Axial length-corneal radius(AL/CR ratio);24.Anterior chamber depth(ACD);25.Lens thickness;26.Parental age at childbirth;27.Grade;28.Presenting visual acuity;29.Negative relative accommodation;30.Positive relative accommodation;31.Age at start of primary school;32.Migrant;33.accommodative convergence to accommodation(AC/A ratio);34.Corneal power;35.Accommodative lag.

*Factors adjusted without results.

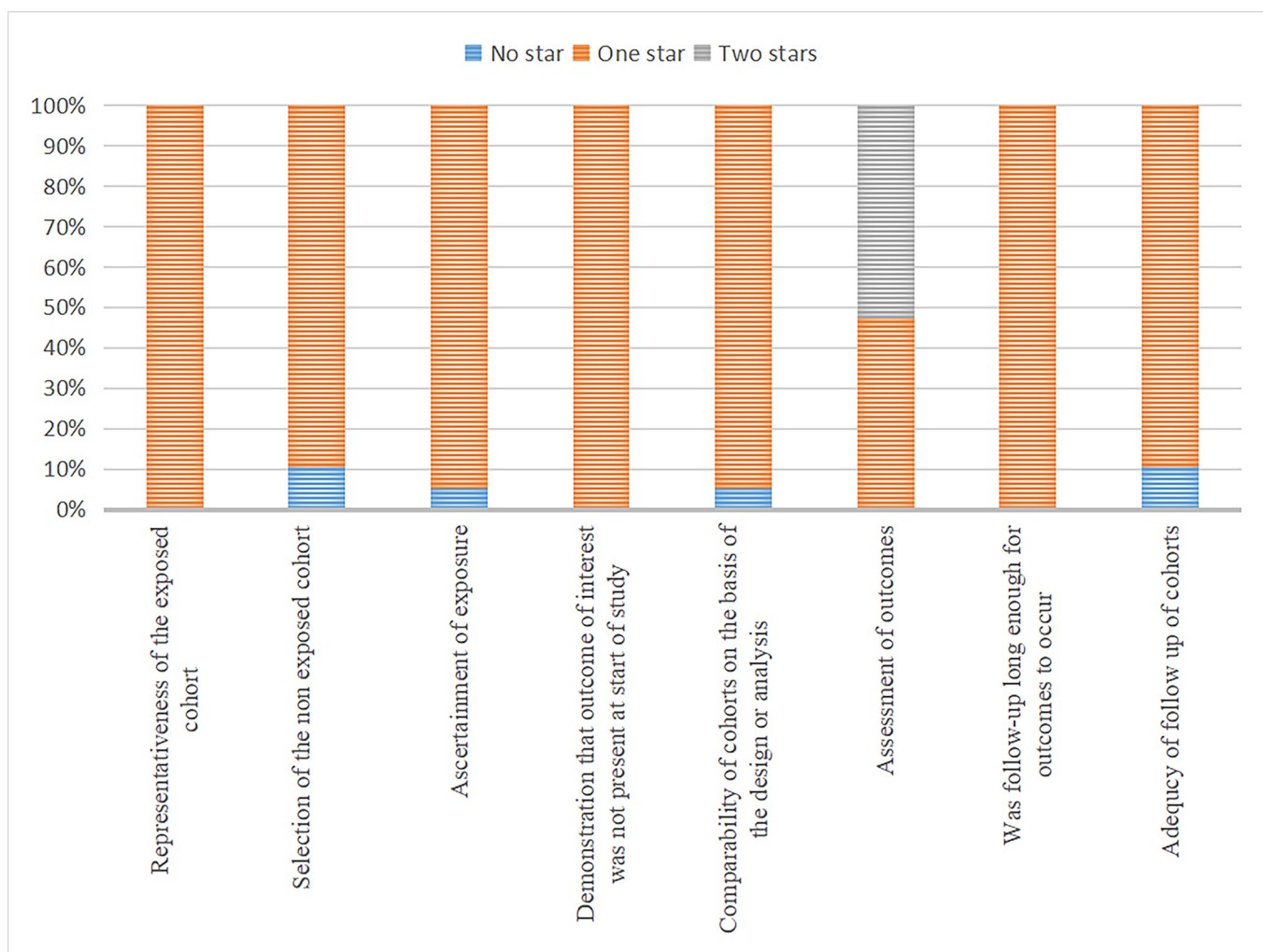

**Fig 2. Risk of bias graph.**

**Table 2. The Negative results of potential factors.**

| Baseline Characteristics | N | OR (95% CI) | Reference |
|---|---|---|---|
| Age | 5080 | 1.06(0.94 to 1.20) | 27,29,31,32 |
| Elementary grade at baseline | | | 29 |
| 2 | 341 | Ref | |
| 3 | 260 | 1.5(0.9 to 2.6) | |
| Age at start of primary school | | | 30 |
| ≤6 years | 101 | Ref | |
| >6 years | 40 | 0.86(0.53 to 1.39) | |
| Migrant | 490 | 0.72(0.38 to 1.37) | 32 |
| Household income, $ | | | 43 |
| ≤2000 | 96 | Ref | |
| 2001 to 5000 | 184 | 1.01 (0.76 to 1.29) | |
| >5000 | 163 | 1.11 (0.84 to 1.45) | |
| Body mass index | 803 | 1.01(0.95 to 1.06) | 27 |
| Region of residence | 1639 | 1.09(0.78 to 1.53) | 31 |
| Height, cm | | | 33 |
| <120.0 (25th Percentile) | 411 | Ref | |
| 120.0–127.5 | 936 | 1.03 (0.87 to 1.22) | |
| >127.5 (75th Percentile) | 384 | 1.02 (0.81 to 1.29) | |
| Weight, kg | | | 33 |
| <20.5 (25th Percentile) | 442 | Ref | |
| 20.5–25.3 | 871 | 1.06 (0.89 to 1.25) | |
| >25.3 (75th Percentile) | 418 | 1.03 (0.83 to 1.28) | |
| AL, mm | | | 33 |
| <22.31 (25th Percentile) | 457 | Ref | |
| 22.31–23.24 | 903 | 1.11 (0.91 to 1.34) | |
| >23.24 (75th Percentile) | 365 | 1.10 (0.82 to 1.48) | |
| Corneal radius of curvature, mm | | | 33 |
| <7.61 (25th Percentile) | 427 | Ref | |
| 7.61–7.95 | 894 | 1.09 (0.90 to 1.32) | |
| >7.95 (75th Percentile) | 407 | 1.13 (0.84 to 1.53) | |
| Negative relative accommodation at baseline, D | 585 | 1.20(1.00 to 1.40) | 33 |
| Accommodative lag, D | 4927 | 0.84 (0.69 to 1.03) | 36 |
| Corneal power, D | 4927 | 1.07(0.97 to 1.19) | 36 |
| Sleep duration, hours | | | 30 |
| ≤49 | 90 | Ref | |
| >49 | 51 | 0.97(0.60 to 1.56) | |
| Outdoor activity time, hours/week | 3666 | | 31,37 |
| ≥9 | | Ref | |
| <4 | | 1.13(0.86 to 1.49) | |
| ≥4 <9 | | 0.80(0.58 to 1.12) | |
| Near work time, hours/day | 1294 | | 27 |
| 0–2.5 | | Ref | |
| 2.5–3.5 | | 1.10(0.82 to 1.48) | |
| >3.5 | | 1.07(0.48 to 1.47) | |
| Near work time, hours/week | 1184 | | 31 |
| <64.5 | | Ref | |
| ≥64.5, <87.5 | | 0.98(0.67 to 1.43) | |

*(Continued)*

**Table 2.** (Continued)

| Baseline Characteristics | N | OR (95% CI) | Reference |
|---|---|---|---|
| ≥87.5 | | 1.00(0.68 to 1.46) | |
| Near work time, hours/week | 1196 | | 39(12 years old cohort) |
| ≤17 | | Ref | |
| <17, ≤25.5 | | 1.43(0.93 to 2.21) | |
| >25.5 | | 1.31(0.83 to 2.06) | |

with cycloplegia: RR:1.27; 95% CI: 1.09–1.44; $I^2$: 22.9%; 7 cohorts; SE ≤ -0.75D with cycloplegia: RR:1.77; 95% CI: 1.38–2.16; $I^2$: 68.0%; 4 cohorts; S2 Fig) might have more possibility of developing myopia.

**Gender factor.** Nine studies [16,18–20,23,24,26,28,34] reported that girls might be 1.28 times more likely to develop myopia than boys (RR: 1.28; 95% CI: 1.22–1.35; $I^2$: 57.0%;

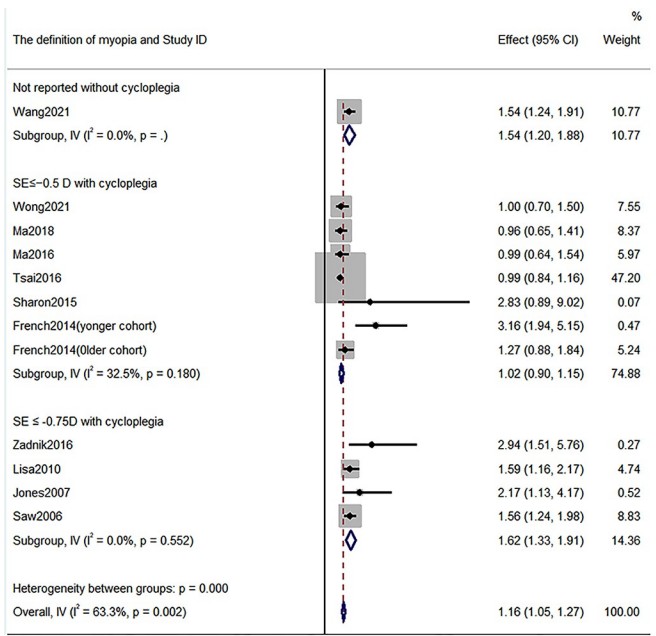

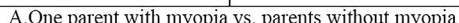

A.One parent with myopia vs. parents without myopia

B.Two parents with myopia vs. parents without myopia

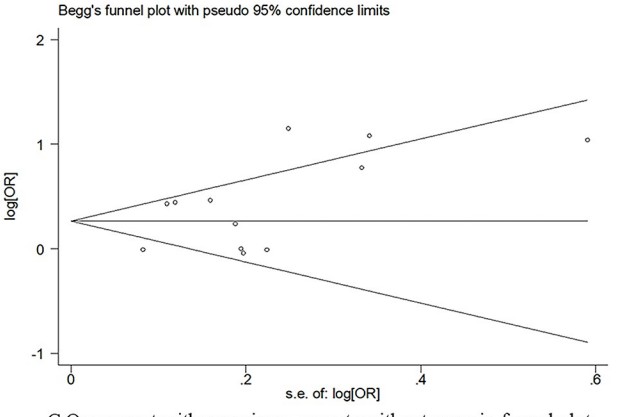

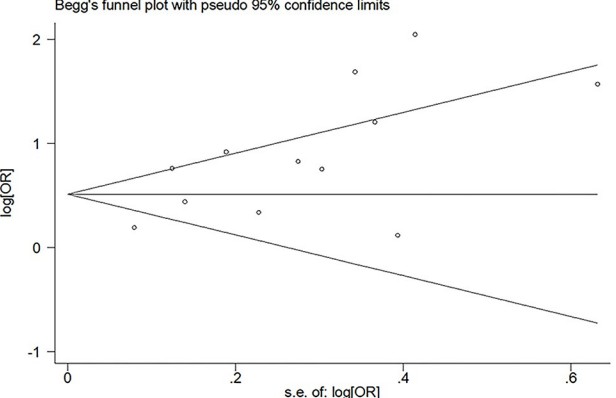

C.One parent with myopia vs. parents without myopia-funnel plot

D.Two parent with myopia vs. parents without myopia-funnel plot

**Fig 3. The funnel plot and forest plot of parent myopia status.**

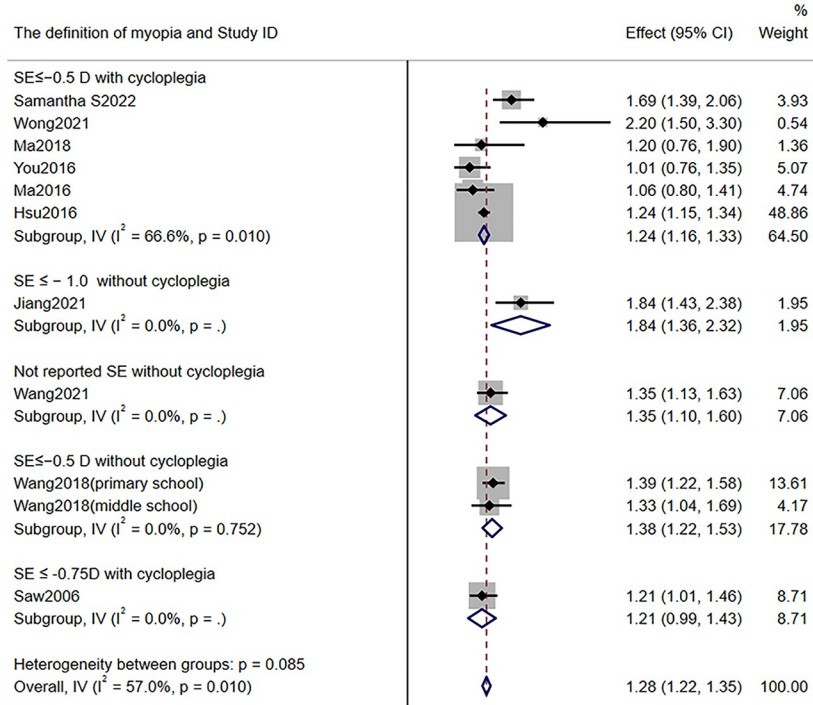

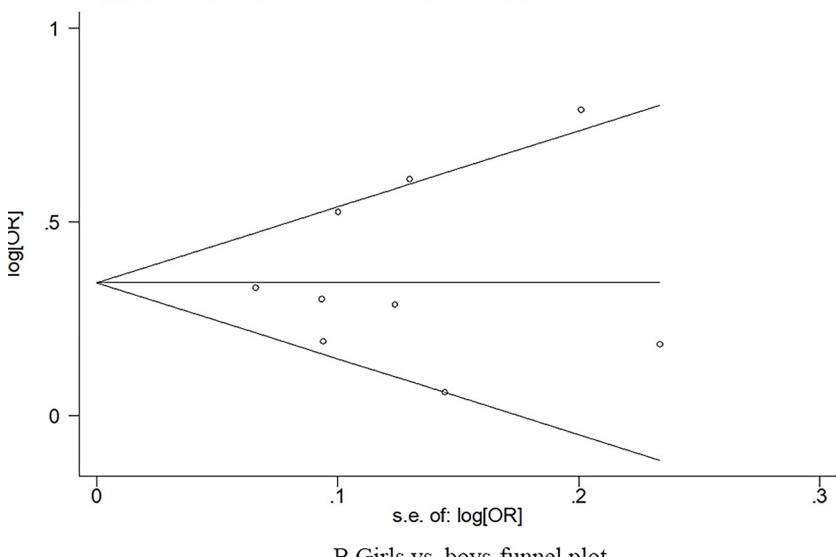

**Fig 4. The funnel plot and forest plot of gender.**

n = 32680; 10 cohorts; middle certainty of evidence; Fig 4). Potential publication bias may not have existed (P = 0.83; Fig 4). One study [32] reported that, among the children (Age: 12), boys are more likely to be myopic.

Six studies[18,20,23,27,32,33] reported the association that longer axial length (RR:1.38; 95% CI: 1.23–1.53; $I^2$: 90.0%; n = 10408; very low certainty of evidence; S1 Fig) and lower SE [18,21,23,27,33] (RR: 2.01; 95% CI: 0.02–0.04; $I^2$: 93.1%; n = 7706; very low certainty of

evidence; S2 Fig). Further, after excluding these two studies(Jiang [18] and Zadnik (First grade) [27]), the heterogeneity of SE diminished significantly (S2 Fig).

**Environmental factors.** The results also showed that a deeper AC [16], a higher AL/CR ratio [16], thinner lenses [16,27], worse presenting visual acuity [20], lower magnitude of positive relative accommodation [20] may be more likely to develop to myopia. Besides, children with astigmatism [27], thicker Crystalline lens [27], and elevated accommodative convergence [27] to accommodation (AC/A ratio) may decrease the risk of myopia onset.

The results of two cohorts [30] demonstrated that East Asians (RR: 5.28; 95% CI: 2.06–13.48; $I^2$ = 87.5%; low certainty of evidence; S3 Fig) might have a greater risk of developing myopia than European Caucasians (RR: 1.99; 95% CI: 1.34–2.96; $I^2$: 23.1%; low certainty of evidence; S3 Fig). A study [34] based on the Raven Standard Progressive Matrices Test suggested that IQ may have an influence on myopia onset (Ref: score 1; score 2; RR: 1.37; 95% CI: 1.08–1.72; Ref: score 1; score 3; RR: 1.50; 95% CI: 1.19–1.89). One study [34] reported that Children living in the suburbs have a lower incidence of myopia(RR: 0.91; 95% CI: 0.83–1.00).

**Behavioural factors.** Eleven cohorts studies [17,19–22,25,28–30,32,33] evaluated outdoor activity time. The results [19,20,30,32,33] of our review demonstrated that longer outdoor activity time could decrease the risk of myopia onset (RR = 0.97; 95% CI: 0.95–0.98; $I^2$: 76.9%; n = 16455; 5 cohort studies; low certainty of evidence; Fig 5). One [17] study showed children who did high-intensity and long-term outdoor exercise may decrease the risk of myopia onset compared with children who played less. On weekdays, myopic children devote more time to outdoor activity time than nonmyopic children (Ref: < 30min; ≥30 min: RR = 0.90; 95% CI: 0.90–0.99) [25]. Two studies reported adequate outdoor activity time could be adapted to reduce the risk of myopia (≥9.33 hours, <14 hours [21], or >22.5 hours [30]).

Overall, the results of four studies [19–21,29] demonstrated that more nine-work time may increase the possibility of myopia onset (RR: 1.05; 95% CI: 1.02–1.07; $I^2$: 22.2%; n = 4689; 4 cohort studies; middle certainty of evidence; Fig 5). French [30] suggested 6 years old children should be careful in avoiding over 19.5 hours nine work time a week. Another three cohorts [18,22,30] reported that there was no evidence that a longer nine work time may lead to a higher risk of myopia onset (Table 2).

S1 Table provides further details of myopia related lifestyles of the included studies.

Compared with 4–5 hours/d of reading, >6 hours/d of reading (RR: 1.65; 95% CI: 1.05–2.57; $I^2$: 68.5%; 3 cohorts; low certainty of the evidence) led to a higher risk of developing myopia [28,30]. Qi [21] reported that compared with the reading distance within 30 cm, the possibility of myopia onset is less if distance over 30 cm (RR: 0.51; 95% CI: 0.27–0.94).

In addition, Chua [26] reported that childhood residential passive smoking, specifically from one spouse, was remarkably connected with an increased possibility of developing myopia. Ma [28] reported a significant difference in myopia among students studying in popular schools compared with those studying in ordinary schools (RR: 2.04; 95% CI: 1.41–2.95).

## Summary of results

Sherwin et al. [35] systematically summarised and analyzed cohort and cross-sectional studies in 2012 to prove the connection between outdoor activity time and myopia onset. Foreman [36] reported the impact of digital smart device use on myopia. Zhang [37] explored the influence of parental myopia on children's myopia in 2015. Huang et al. [38] suggested that more near work time may increase the incidence. Xiong [39] further explored the influence of outdoor activities on myopia progression. Lanca [40] pooled the results of six cohort studies and found that screen time (OR = 1.09; 95% CI: 0.96–1.08) may not affect myopia. Pirro et al. [41] performed meta-analysis of genome-wide. Previous systematic reviews have mainly focused

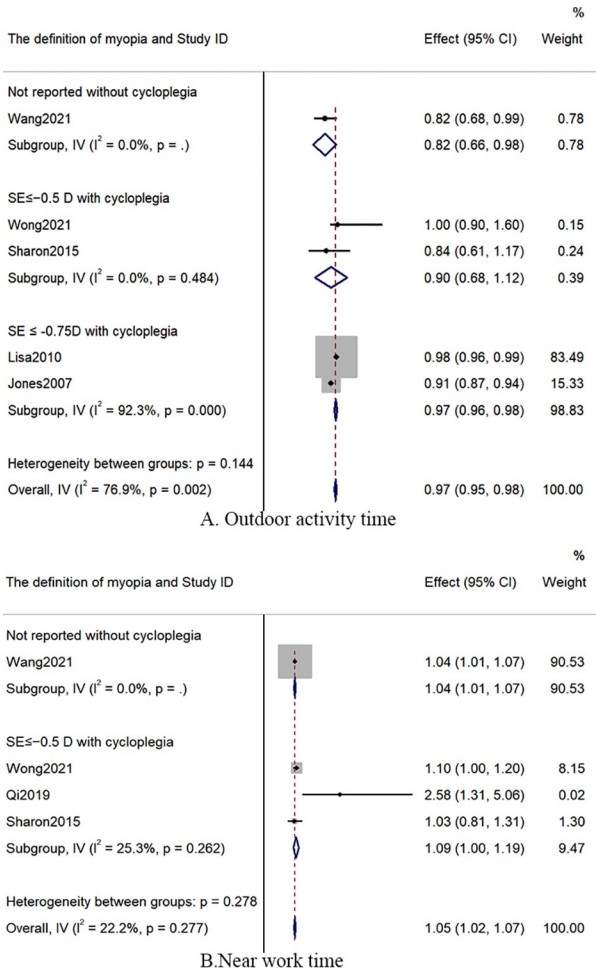

Fig 5. The forest plot of outdoor activity time and near work time.

on myopia-related lifestyles. This study was based on a rigorous search of 19 full-text articles and an evaluation of the included studies, with a score of 6–9. The underlying factors for myopia are mixed, including positive (20) and negative (17) results. The onset of myopia is related to environmental factors, gender, parental myopia, and various eye indicators. The burden of myopia is higher in low-income countries than in developed countries. Lifestyle factors associated with myopia are associated with the onset of myopia. Interestingly, children with high IQs had an increased risk of myopia. Analyzing gender and age, girls were more likely to develop myopia, which can occur at any age, from early childhood to late adulthood. This suggests that we need to focus on girls when it comes to child vision protection. In terms of genetic factors, children of parents with myopia are more likely to develop myopia, indicating that genetic factors of myopia affect children's vision. It needs to strengthen research and understanding of genetic factors to develop more targeted vision protection strategies. In terms of environmental factors, spending more time outdoors and less time working in close quarters appeared to significantly reduce the incidence of myopia. This suggests that children should be encouraged to spend more time outdoors and less time doing close visual work. Visual acuity related factors: In children, lower SE, longer AL, lower positive relative regulation, poor visual acuity, deeper anterior chamber, and thinner lens may be associated with the onset of myopia. These factors provide guidance for the early prevention of myopia. In

addition, living in the suburbs may be a protective factor for myopia. The burden of myopia in poor countries is higher than that in developed countries in terms of geographical factors. This shows that we need to strengthen support and assistance for children's vision protection in poor areas. The quality of evidence for the factors assessed was medium to low or very low. This suggests that further high-quality research is needed to gain a more comprehensive and in-depth understanding of the risk factors for myopia and to develop more effective preventive measures.

**Implications for practice.**   The increased possibility of myopia in children with myopic parents showed that the onset of myopia is highly heritable. Clinicians and parents should care more for children with longer AL, lower SE, the lower magnitude of positive relative accommodation, a higher AL/CR ratio, thinner lenses, a lower magnitude of positive relative accommodation, worse presenting visual acuity, elevated AC/A ratio, deeper anterior chamber, or thinner crystalline lens. Girls are more prone to develop myopia compared to boys. Owing to some socioeconomic issues, the risk of myopia onset in underprivileged countries is far greater than that in developed countries. Correspondingly, the increased prevalence of myopia onset may also create a financial burden on society as a whole. In East Asia, the traditional culture and education model that do not encourage children to spend time outdoors aggravate the occurrence of myopia. Policymakers should pay more attention to potentially vulnerable groups who have higher chances of developing myopia.

Adequate outdoor activity time and reasonable near-work time should be adjusted to prevent myopia. A shorter reading distance and longer reading time may increase the risk of myopia onset. Decreasing near-work time and improving near posture (over 30 cm reading distance) through proper desk and chair adjustments could slow the onset of myopia.

Age, household income, sleep duration, accommodative lag, corneal power, etc., may not be associated with myopia onset. Myopia can occur at any age, from early childhood to late adulthood. In the clinical scenario, both the positive and negative results should be inferred in the context because neither false-positive nor false-negative results are rare.

**Implications for research.**   The genes expressed differentially in chick form-deprivation myopia indicates prospective mechanistic differences concerning myopia onset and advancement of established myopia [42]. Research that independently explores potential factors for myopia onset should be designed. Given that myopia has complex aetiologies, the effects of some risk factors, such as genes, sleep duration, and light exposure, remain controversial and lack sufficient cohort studies for confirmation. We want to investigate the effects of genes, light exposure time, and sleep duration in future studies owing to insufficient studies in this regard. Further studies should be designed to explore the differences in myopia onset between low- and middle-income countries and developed countries. Age in different studies was reported according to the same criteria (range or mean ± standard deviation). Therefore, the representativeness of the study population should be considered. The baseline of the cycloplegic SE for every participant was provided. Exploring the relationship between myopia and parental refractive status should not simply include the number of parents with myopia. Urbanization, as a critical risk factor for myopia, should be paid attention to by more and more cohort studies in the future. Researchers should comprehensively explore the influence of parental refractive diopters on paediatric myopia. Outdoor activity time and near-work time should be considered with unified standards when the study is designed. Myopia-related lifestyles have only been proven to be related to myopia by current cohort studies. Randomized controlled trials should be designed to explore recommendations that are more conducive to guiding practice to change children's lifestyles.

**Strengths and limitations.**   Whilst there is no comprehensively recognized scheme for the prevention of myopia onset, it is very crucial to identify the modifiable risk factors that are

connected with its onset. This is the frontmost research that evaluated all potential factors with positive or negative results. The potential factors with negative results can be used to further optimize future therapies and direct areas of future research. Previous meta-analyses of risk factors for myopia used cross-sectional studies to extensively discuss the relationship between risk factors and myopia progression of myopic children by analyzing cross-sectional studies. The analyses of cross-sectional studies cannot establish causality between risk factors and myopia. Therefore, we explored the relationship between potential factors and myopia onset by analyzing cohort studies, which can provide a greater understanding of causality. This study helps to analyze the risk of myopia in children in real situations, and combines it with clinical factors to establish a predictive model. Myopia prevention research can identify and recruit children with high risk of myopia based on these risk factors. The evidence quality of the outcomes was evaluated by GRADE. Most of the studies were assessed as high quality by NOS. According to the requirements of Cochrane, we included the research related to myopia onset as comprehensively as possible to minimize the influence of publication bias, and the large sample size ensured the reliability of our results. However, this review had several limitations. First, a relatively large heterogeneity was evident in our review. The subgroups and sensitivity analyses were used to identify the relatively large heterogeneity. There was a high degree of heterogeneity due to sample diversity, mean age of participants, differences in definitions of myopia and myopia-related lifestyles, and multiple confounding factors. The degree of correction for potential confounders varies considerably across studies, and residual confounding can have some impact on the assessment of individual outcomes and the overall OR. The findings of existing studies indicated significant heterogeneity, and the quality of the evidence was generally low. Second, different outdoor activity standards limit the integration of data. Third, the insufficient number of studies limits the evidence strength of eye parameters. The study provides important guidance for future prevention and management of myopia in children. Future studies could further explore the influence of other factors such as positive relative regulation degree, corneal curvature, and corneal thickness on myopia in children, and further investigate the role of outdoor activities and close working time. In addition, future research could further explore the interaction between environmental and genetic factors to better understand the pathogenesis of childhood myopia. Ultimately, the results of this work could provide a scientific basis for the development and implementation of public health policies to reduce the health and economic burden of myopia.

## Conclusions

According to the analysis, girls were more likely to develop myopia; Children whose parents are near sighted are more likely to develop myopia; More time outdoors and less time spent working in close quarters may significantly reduce the incidence of myopia. Children with lower SE, longer AL, lower positive relative regulation, poor vision, deeper anterior chamber, and thinner lens may be related to the onset of myopia. In addition, the burden of myopia is higher in poor countries than in developed ones. Therefore, according to these risk factors, corresponding prevention and intervention measures should be taken to reduce the incidence and burden of myopia in children. Myopia prevention strategies should be designed based on environmental factors, gender, parental myopia, and eye indicators to explore more beneficial to the eye health lifestyle of children.

## Supporting information

**S1 Checklist. PRISMA 2020 checklist.**
(DOCX)

**S1 Fig. The sensitivity analysis and forest plot of axial length.**
(PNG)

**S2 Fig. The sensitivity analysis and forest plot of cycloplegic spherical equivalent degree.**
(PNG)

**S3 Fig. The forest plot of ethnicity.**
(PNG)

**S1 Table. The definition and usage of outdoors activities and near work.**
(DOCX)

**S2 Table. The risk of bias assessment for each study according to Newcastle-Ottawa quality assessment scale.**
(DOCX)

**S1 File.**
(DOCX)

**S1 Data.**
(ZIP)

## Acknowledgments

We thank Elsevier Language Editing Services and Mr Haroon Mujahid for assistance with language editing (https://webshop.elsevier.com/language-editing-services/language-editing/).

## Author Contributions

**Conceptualization:** Mingkun Yu, Mei Han, Hongsheng Bi.

**Data curation:** Mingkun Yu, Yuanyuan Hu.

**Formal analysis:** Mingkun Yu.

**Funding acquisition:** Hongsheng Bi.

**Investigation:** Yuanyuan Hu.

**Project administration:** Jiawei Song, Ziyun Wu, Zihang Xu, Yi Liu, Zhen Shao, Guoyong Liu, Zhipeng Yang.

**Resources:** Jiawei Song, Ziyun Wu, Zihang Xu, Yi Liu, Zhen Shao, Guoyong Liu, Zhipeng Yang.

**Software:** Mei Han, Jiawei Song, Ziyun Wu, Zihang Xu, Yi Liu, Zhen Shao, Guoyong Liu, Zhipeng Yang.

**Supervision:** Yuanyuan Hu, Mei Han, Hongsheng Bi.

**Validation:** Hongsheng Bi.

**Visualization:** Yuanyuan Hu, Mei Han, Hongsheng Bi.

**Writing – original draft:** Mingkun Yu, Hongsheng Bi.

**Writing – review & editing:** Mingkun Yu, Yuanyuan Hu, Mei Han, Hongsheng Bi.

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
