## [Decision Letter · Decision Letter 0]

3 Jul 2023

PONE-D-23-13955Global risk factor analysis of myopia onset in children: A systematic review and meta-analysisPLOS ONE

Dear Dr. Bi,

Thank you for submitting your manuscript to PLOS ONE. Authors has addressed most comments in the previous submission and made significant improvement in this version. But, some mistakes should be corrected before the further decision. After careful consideration, we feel that it has merit but does not fully meet PLOS ONE’s publication criteria as it currently stands. Therefore, we invite you to submit a revised version of the manuscript that addresses the points raised during the review process.

We look forward to receiving your revised manuscript.

Kind regards,

Der-Chong Tsai, MD, PhD

Academic Editor

PLOS ONE

4. Please include your tables as part of your main manuscript and remove the individual files. Please note that supplementary tables should be reamin as separate "supporting information" files"

Reviewers' comments:

Reviewer's Responses to Questions

**Comments to the Author**

1. Is the manuscript technically sound, and do the data support the conclusions?

Reviewer #1: Yes

Reviewer #2: Yes

2. Has the statistical analysis been performed appropriately and rigorously? 

Reviewer #1: Yes

Reviewer #2: Yes

3. Have the authors made all data underlying the findings in their manuscript fully available?

Reviewer #1: Yes

Reviewer #2: Yes

4. Is the manuscript presented in an intelligible fashion and written in standard English?

Reviewer #1: Yes

Reviewer #2: Yes

5. Review Comments to the Author

Reviewer #1: This meta-analysis aimed to comprehensively assess the risk factors affecting myopia in children to develop more effective prevention and treatment strategies.

The authors committed to analyze the risk and preventing factors for myopia onset. Both methods and discussions are sensible. As myopia is a global problem and becoming more and more important, this article is pivotal for ophthalmologist and public health practitioner.

One miss speaking should be corrected: page 19, line 397. ”clo-e” to ”close”

Reviewer #2: Although the authors had addressed most of my suggestions I raised in first review, some points as shown below still need to be clarified.

Major:

1. Some study such as PMID: 26802174 had shown children from urban environments have 2.6 times the odds of myopia compared with those from rural environments. Thus, urbanization plays an important role in myopia development. However, this study did not address this key risk factor.

2. I suggest to adopt risk of bias assessment for each study according to NOS so that the readers could understand the quality of each study.

Minor:

1. The term ”multivariate” should be corrected to “multiple” or “multivariable” in Table 1.

2. The term “exp(b)” should be corrected to “OR” or “Odds Ratio” in appendix figure 3.

6. PLOS authors have the option to publish the peer review history of their article (what does this mean?). If published, this will include your full peer review and any attached files.

Reviewer #1: No

Reviewer #2: No

---

## [Author Response · Author response to Decision Letter 0]

3 Aug 2023

Dear Editor:

We would like to resubmit the revised manuscript entitled ‘Global risk factor analysis of myopia onset in children: A systematic review and meta-analysis’ for consideration. We would like to thank the reviewers for thoroughly reviewing our manuscript and making many thoughtful comments. We have added significant new data, described in detail below, and revised the manuscript to address reviewers’ comments. Here are our point-by-point responses:

Reviewer 1:This meta-analysis aimed to comprehensively assess the risk factors affecting myopia in children to develop more effective prevention and treatment strategies.

The authors committed to analyze the risk and preventing factors for myopia onset. Both methods and discussions are sensible. As myopia is a global problem and becoming more and more important, this article is pivotal for ophthalmologist and public health practitioner.

1. One miss speaking should be corrected: page 19, line 397. ”clo-e” to ”close”.

Response: Thanks for your valuable comments. We corrected the sentence.

Reviewer #2: Although the authors had addressed most of my suggestions I raised in first review, some points as shown below still need to be clarified.

Major:

1. Some study such as PMID: 26802174 had shown children from urban environments have 2.6 times the odds of myopia compared with those from rural environments. Thus, urbanization plays an important role in myopia development. However, this study did not address this key risk factor.

Response: The questions and suggestions raised by you are essential and helpful, which inspired us to optimize our findings further. Unfortunately, we previously overlooked this key risk factor. We supplemented the results and discussion that Tsai DC et al. mentioned as the key risk factor in the occurrence of myopia in 2016. 

Results(Page 12, Line 244-245):

One study34 reported that Children living in the suburbs have a lower incidence of myopia(RR: 0.91；95% CI: 0.83-1.00).

Discussion

In addition, living in the suburbs may be a protective factor for myopia.(Page 14, Line 306-307). 

Urbanization, as a critical risk factor for myopia, should be paid attention to by more and more cohort studies in the future.(Page 16-17, Line 353-354)

In addition, we carefully explored the differences from Rudnicka AR et al.'s research. Our research mainly focuses on the incidence of myopia, while their research focuses on the prevalence of myopia. Only cohort studies are included in our research, so few articles explore this key factor.

2. I suggest to adopt risk of bias assessment for each study according to NOS so that the readers could understand the quality of each study.

Response: Thank you for your constructive comments, which helped us improve the manuscript. We designed appendix table 2 to provide the risk of bias assessment for each study to help readers understand the quality of each study.

Minor:

1. The term ”multivariate” should be corrected to “multiple” or “multivariable” in Table 1.

Response: Thanks for your valuable comments. We corrected the term.

2. The term “exp(b)” should be corrected to “OR” or “Odds Ratio” in appendix figure 3.

Response: Thanks for your valuable comments. We corrected the term. In this revision, we have corrected the appendix figure 3 and checked the manuscript carefully, including each picture.

---

## [Decision Letter · Decision Letter 1]

30 Aug 2023

Global risk factor analysis of myopia onset in children: A systematic review and meta-analysis

PONE-D-23-13955R1

Dear Dr. Bi,

We’re pleased to inform you that your manuscript has been judged scientifically suitable for publication and will be formally accepted for publication once it meets all outstanding technical requirements.

Kind regards,

Der-Chong Tsai, MD, PhD

Academic Editor

PLOS ONE

Additional Editor Comments (optional):

Reviewers' comments:

Reviewer's Responses to Questions

**Comments to the Author**

1. If the authors have adequately addressed your comments raised in a previous round of review and you feel that this manuscript is now acceptable for publication, you may indicate that here to bypass the “Comments to the Author” section, enter your conflict of interest statement in the “Confidential to Editor” section, and submit your "Accept" recommendation.

Reviewer #2: All comments have been addressed

2. Is the manuscript technically sound, and do the data support the conclusions?

Reviewer #2: Yes

3. Has the statistical analysis been performed appropriately and rigorously? 

Reviewer #2: Yes

4. Have the authors made all data underlying the findings in their manuscript fully available?

Reviewer #2: Yes

5. Is the manuscript presented in an intelligible fashion and written in standard English?

Reviewer #2: Yes

6. Review Comments to the Author

Reviewer #2: The authors had perfprmed the scientific research with data that supports the conclusions. No additional comment

7. PLOS authors have the option to publish the peer review history of their article (what does this mean?). If published, this will include your full peer review and any attached files.

Reviewer #2: No

---

## [Editor Report · Acceptance letter]

10 Sep 2023

PONE-D-23-13955R1 

Global risk factor analysis of myopia onset in children: A systematic review and meta-analysis 

Dear Dr. Bi:

I'm pleased to inform you that your manuscript has been deemed suitable for publication in PLOS ONE. Congratulations! Your manuscript is now with our production department. 

Kind regards, 

on behalf of

Dr. Der-Chong Tsai 

Academic Editor

PLOS ONE